# Eosinophils in Oral Disease: A Narrative Review

**DOI:** 10.3390/ijms25084373

**Published:** 2024-04-16

**Authors:** Huda Moutaz Asmael Al-Azzawi, Rita Paolini, Nicola Cirillo, Lorraine Ann O’Reilly, Ilaria Mormile, Caroline Moore, Tami Yap, Antonio Celentano

**Affiliations:** 1Melbourne Dental School, The University of Melbourne, 720 Swanston Street, Carlton, VIC 3053, Australia; hasmael@student.unimelb.edu.au (H.M.A.A.-A.); rita.paolini@unimelb.edu.au (R.P.); nicola.cirillo@unimelb.edu.au (N.C.); moore@unimelb.edu.au (C.M.); tspyap@unimelb.edu.au (T.Y.); 2The Walter and Eliza Hall Institute of Medical Research, 1G Royal Parade, Parkville, VIC 3052, Australia; oreilly@wehi.edu.au; 3Department of Medical Biology, University of Melbourne, Parkville, VIC 3010, Australia; 4Department of Translational Medical Sciences, University of Naples Federico II, 80131 Naples, Italy; ilaria.mormile@unina.it

**Keywords:** eosinophils, oral diseases, oral cancer, oral potentially malignant disorders

## Abstract

The prevalence of diseases characterised by eosinophilia is on the rise, emphasising the importance of understanding the role of eosinophils in these conditions. Eosinophils are a subset of granulocytes that contribute to the body’s defence against bacterial, viral, and parasitic infections, but they are also implicated in haemostatic processes, including immunoregulation and allergic reactions. They contain cytoplasmic granules which can be selectively mobilised and secrete specific proteins, including chemokines, cytokines, enzymes, extracellular matrix, and growth factors. There are multiple biological and emerging functions of these specialised immune cells, including cancer surveillance, tissue remodelling and development. Several oral diseases, including oral cancer, are associated with either tissue or blood eosinophilia; however, their exact mechanism of action in the pathogenesis of these diseases remains unclear. This review presents a comprehensive synopsis of the most recent literature for both clinicians and scientists in relation to eosinophils and oral diseases and reveals a significant knowledge gap in this area of research.

## 1. Introduction

Eosinophils are bone marrow-derived granulocytes, along with neutrophils and monocytes, which reside in blood and tissues (Figure 1).

First discovered in amphibians (e.g., frogs) and mammals (e.g., dogs, rabbits and humans) in 1879 by Ehrlich [1], their expression in a wide variety of species suggests an evolutionarily conserved role. However, their role in immunity remains to be clarified in many disease situations. Eosinophils constitute 1–3% of circulating blood leukocytes [2], with a blood half-life of 8–18 h, but can survive for several weeks in the tissues [3]. As mature cells in the periphery, eosinophils are characterised by a copious basophilic cytoplasm with numerous coarse granules containing cationic proteins; the major basic protein, eosinophil cationic protein, is an eosinophil-derived neurotoxin and eosinophil peroxidase [4]. When stimulated, eosinophilic granules are released from eosinophils, where they act as the first line of immune defence against infectious agents such as microbes, parasites, and allergens are associated with blood and solid cancers [5,6]. While providing a defensive immune response role against viral, bacterial and helminth pathogens [7], simultaneously, eosinophils also play a role in tissue destruction by initiating inflammation through the release of eosinophil-derived cytotoxic mediators [8], with a particularly detrimental role to play in allergic disorders [2].

**Figure 1 ijms-25-04373-f001:**
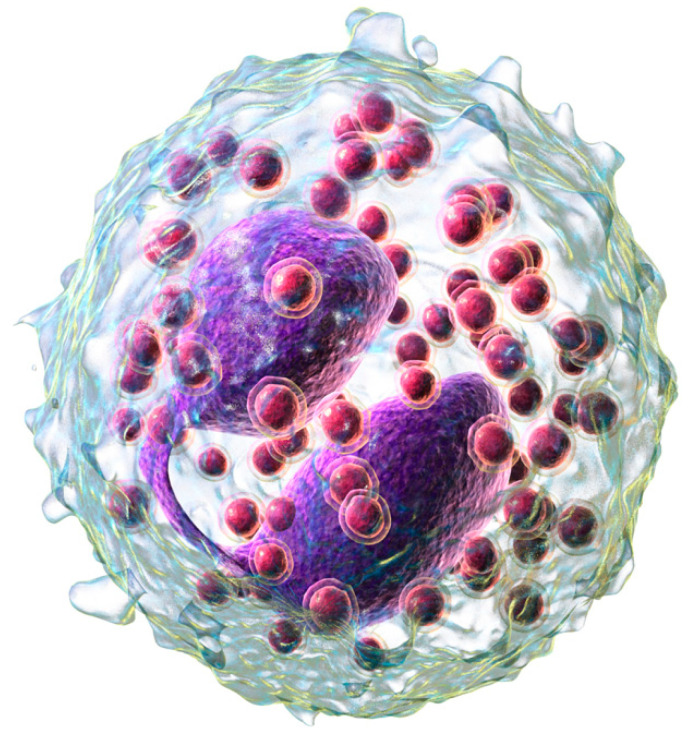
A 3D rendering of eosinophil. From [9].

The peripheral blood absolute eosinophil count (AEC) normal value ranges from 0.05 to 0.5 × 10^9^/L, while reference values for mature eosinophils in bone marrow aspirates are between 1% and 6% [10]. According to the classification of eosinophilic disorders proposed in 2011 by the International Cooperative Working Group on Eosinophil Disorders (ICOG-EO) and revised in 2022, blood eosinophilia is defined by an AEC above 0.5 × 10^9^/L, while hypereosinophilia (HE) necessitates an AEC of ≥1.5 × 10^9^/L. Persistent HE may be associated with eosinophil infiltration into the tissues, leading to tissue and organ damage caused mainly by the release of eosinophil effector molecules [10,11]. Moreover, tissue eosinophilia (TE) requires histopathological demonstration [3].

Eosinophils develop in the bone marrow from myeloid hematopoietic progenitor cells, with several critical cytokines, including interleukin (IL)-5, IL-3, and the granulocyte-macrophage colony-stimulating factor (GM-CSF) involved in their development and differentiation [7]. Upon release into the circulation and terminal differentiation, eosinophils are recruited in various tissues by chemokines, such as the eosinophil chemoattractant eotaxin [12]. Eosinophilia is a characteristic feature of helminth infection, allergies such as asthma, and a wide spectrum of diseases [7]. Eosinophils have multiple functions, including antigen recognition in viral, bacterial, and parasitic infections, thereby playing a role in the innate immune response. They also contribute to the secretion of cytokines in the case of acute and chronic inflammation and are important for tissue health and remodelling [13]. Upon activation, eosinophils have the capability of rapidly releasing a battery of immunomodulatory factors, including over 35 cytokines (e.g., IL-1a, -2, -3,-4, -5, -6, -10, -11, -12, -13, -16, -25), growth factors (HB-EGF, NGF, PDGF-b, SCF, TCFa, TGFb and VEFG), chemokines (CCL-3, -5, -5, -11,-17,-22,-23 and CXCL-1, -5, -8, -9, -10, -11), and others (GM-CSF, IFNg, TNF), which unlike T and B cells, occurs with minutes [14].

There are several recent excellent reviews outlining the regulatory and cellular function of eosinophils and their role in health and disease [15,16]. Pertinent to this review, excess eosinophils in mucosal biopsies of one or multiple gastrointestinal (GI) tract sites play a causative role in eosinophilic GI disorders (EGIDs). For example, eosinophilic esophagitis (EoE), eosinophilic gastritis (EG), and eosinophilic gastroenteritis (EGE) are groups of chronic immune-mediated diseases characterised by histopathologic eosinophilic infiltration in the oesophagus or stomach, often associated with an abnormal response to dietary antigens or parasitic infection. Given this significant involvement in gastrointestinal diseases [17,18,19], these studies are suggestive of the potential role of eosinophils throughout the GI tract, including the oral cavity. Eosinophils are normally present within oral mucosa as individual cells; however, in response to allergic reactions or parasitic infections, the infiltration of eosinophils may occur [20]. It is also well documented that eosinophil infiltration is associated with tumour formation (tumour-associated tissue eosinophilia (TATE)). However, the underlying mechanisms behind this phenomenon remain poorly understood [6,21].

To the best of the author’s knowledge, this is the first review to comprehensively summarise the role of eosinophils, specifically in oral diseases. This narrative review was conducted through a selective literature search to consolidate findings from relevant studies (Figure 2)

We present the current knowledge base in this subject area to provide a more comprehensive understanding of the relationship between eosinophils and oral diseases, which have been stratified here into two main categories:Oral diseases—associated with TE and/or blood eosinophilia (BE).Systemic diseases—associated with both eosinophilia and oral lesions.

These are followed by two additional paragraphs dedicated to exploring the role of eosinophils in oral precancerous lesions and oral cancer.

While our initial emphasis on precancerous and cancerous aspects may have been pronounced, we recognise that confining our entire review to these dimensions diverges from our original intention.

The aim of this study is to provide a comprehensive overview of the role of eosinophils in oral diseases, including their involvement in tissue eosinophilia, blood eosinophilia, and systemic diseases with oral manifestations, aiming to enhance understanding among clinicians and pathologists and to serve as a reference for diagnosis and management in oral medicine.

Our deliberate highlighting aimed to stimulate discourse on the debated link between eosinophils and these conditions. Additionally, the vast array of pathological entities within cancer and precancer necessitated increased space dedication compared to other oral/systemic disorders.

## 2. Eosinophils and Oral Diseases

### 2.1. Oral Diseases Associated with Tissue Eosinophilia and/or Blood Eosinophilia 

This category involves a group of oral diseases characterised by the presence of TE in a histopathological examination and/or a presence that may be concurrently associated with BE: 

#### 2.1.1. Reactive Lesions

An **oral eosinophilic ulcer** is an inflammatory reactive lesion of unknown aetiology, with trauma implicated as causative; hence, it is not commonly observed clinically. Various terminologies have been used to describe this lesion, including “eosinophilic ulcer”, “eosinophilic granuloma of soft tissue”, and “traumatic ulcerative granuloma with stromal eosinophilia (TUGSE)” [22]. It was first described by Popoff in 1956 and later by Shapiro and Juhlin [23]. Clinically, TUGSE manifests as a painful single ulcer with indurated borders, mostly on the surface of the tongue, but can involve any oral mucosal site and is a self-limiting lesion that tends to heal spontaneously (Figure 3) [23,24].

Due to its clinical presentation and prolonged duration, TUGUE should not be neglected as it closely resembles OSCC [25]. A histopathological examination of these ulcerative lesions reveals that underlying connective tissue deep into muscle bundles are chronically infiltrated with inflammatory cells such as neutrophils and lymphocytes, with eosinophils frequently observed as part of a mixed infiltration [26]. Inflammatory cytokines and chemokines released from eosinophils have been suggested to play a role in the pathogenesis of this entity; however, the exact mechanism is still unclear [27].

**Riga–Fede disease:** this eponym is used to describe an infantile benign condition characterised by the presence of ulcers on the dorsal surface of the tongue (60% of lesions), the lips, palate, mucous membrane of the vestibule and the floor of the mouth. The disease is closely related to TUGSE but is predominantly observed in infants, typically caused by repetitive trauma to the oral mucosa from natal or neonatal teeth [28]. This traumatic and often lingual ulceration is accompanied by eosinophilic granuloma and the eosinophilic ulceration of the tongue and oral mucosa [29]. Ulcerations may be painful, and extraction or odontoplasty are the most effective treatments [28]. Due to the distinctive clinical features, diagnostic assessment and specific treatment, it is useful to classify Riga–Fede disease and eosinophilic ulcers separately from a clinical standpoint [30].**Eosinophilic granuloma** is a rare benign bone lesion, accounting for less than 1% of bone tumours. It mostly affects children under 10 years of age, with the mandible being the most frequently affected site. It represents a mild localised type of oral Langerhans cell histiocytosis (LCH) without malignant transformation. Radiologically, it manifests as teeth resorption, with the teeth appearing to be floating in the air. A histopathological examination reveals the scattered sheets of eosinophilic infiltrates [31,32].

#### 2.1.2. Oral Lichenoid Lesions (OLLs)

OLLs are a group of oral lesions associated with specific initiation triggers, including oral lichenoid drug reactions (OLDRs), oral lichenoid contact lesions (OLCLs) and lesions associated with graft versus host disease (GvHD). OLLs usually occur unilaterally and in less established sites such as the gingiva, lip and palate, with a topographical distribution following a causative trigger [33]. Histopathologically, these lesions are characterised by inflammatory infiltrates composed of plasma cells and also eosinophils [34]. In most cases, eosinophils were located in the superficial lamina propria or between the epithelium–lamina propria interface or the superficial lamina propria [34]. This category also involves oral lichenoid reactions (OLRs) to dental materials, particularly allergic reactions to amalgam [35]. OLRs manifest clinically with a range of presentations, from asymptomatic striae and plaque-like lesions to painful ulcerative and erythematous lesions (Figure 4 and Figure 5) [36].

#### 2.1.3. Oral Vesicular and Bullous Lesions

**Pemphigus vegetans:** this is the rarest clinical variant of pemphigus, comprising 1–2% of all pemphigus cases. It differentiates from pemphigus vulgaris through the development of vegetative plaques in intertriginous regions and oral mucosa and by the presence of autoantibodies against desmoglein 3. Two clinical subtypes exist characterised by flaccid bullae and erosions (Neumann subtype) or pustules (Hallopeau subtype). Both subtypes progress to the development of hyperpigmented vegetative plaques accompanied by pustules. A histopathological examination reveals intraepithelial abscesses with the significant presence of eosinophils [36].**Bullous pemphigoid (BP)** is a chronic autoimmune blistering disease which mainly involves the skin but, on rare occasions, can affect the oral mucous membrane (Figure 6), usually associated with an older age demographic [37]. It is a subtype of the autoimmune blistering disease (AIBD), in contrast to the mucous membrane pemphigoid (MMP), which is also a subtype of AIBD but with a predominantly oral mucous presentation. Mucosal lesions of BP are mild, affecting only about 10–20% of patients, and exhibit less aggressive mucosal erosion and blistering lesions [38,39]. A histopathological examination is characterised by submucosal blister formation with a mixed inflammatory cell infiltrate, particularly eosinophils. BE has also been distinguished as a characteristic of bullous pemphigoid [40,41]. Peripheral eosinophilia is associated with increased levels of IL-5, the key mediator of eosinophil activation produced by T helper cells and mast cells [42]. Elevated levels of IL-5 in serum and blister fluids have been reported in 50–60% of bullous pemphigoid (BP) cases [43]. Moreover, eotaxin-1/CCL11, a chemokine also known as eosinophil chemotactic protein, is overexpressed by keratinocytes within the BP blisters, which may instigate the recruitment of eosinophils into the blister site [44,45,46].

### 2.2. Systemic Diseases Associated with Oral Lesions and Eosinophilia

This category involves a group of systemic diseases that have several oral manifestations and are associated with peripheral BE; moreover, they may be concurrently associated with TE, as shown in Table 1:**Crohn’s disease (CD)** is a chronic inflammatory bowel disease that affects the GI tract. Oral manifestations of CD (oral Crohn’s) are common and include ulcers, the fissuring of the lip, cobblestoning of the buccal mucosa, angular cheilitis, mucosal polyps and perioral erythema (Figure 7). A blood investigation into CD patients revealed leucocytosis and eosinophilia, with eosinophils identified as important contributing cells to tissue immune cell infiltration [47,48].**Kimura’s disease** is a rare chronic inflammatory disease associated with the systematic elevation of eosinophils levels. First described by Kimura et al. [49], it presents as a non-tender subcutaneous lesion in the head and neck region, with only a few reports in the oral cavity [50]. Histopathological examination indicates a nodular architecture and follicular hyperplasia with intense eosinophilic infiltration [51,52,53].**Hypereosinophilic syndrome (HES)** is a rare disorder characterised by a persistent absolute eosinophil count ≥ 1.5 × 10^9^/L, whereby “persistent” denotes the detection of HE on at least two measurements with a minimum of a 2-week interval between them, together with the documentation of eosinophilic infiltrates or the presence of their products, eosinophil-associated organ damage, and the exclusion of another underlying disorder as the primary driver of organ damage [11] The oral presentation of HES manifests as painful mucosal lesions presenting as ulcerations and erosions localized to the lips, gingiva, tongue, and palatal mucosa, all associated with intense eosinophilic infiltration [54].**IgG4-related disease** is a systemic immune-mediated condition characterised by an elevation in serum IgG4 and IgE levels and eosinophilia [55,56]. It can affect any part of the body and is characterised by pancreatitis, retroperitoneal fibrosis, mucosal and glandular infiltration, as well as cutaneous manifestations [36]. The salivary glands are the third most affected tissues after the pancreas and biliary tract [57], mainly resulting in IgG4-related Mikulicz Diseases (IgG4-MD) and IgG4-related Chronic Sclerosing Sialadentis (IgG4-CSS) [58]. In addition, the tongue and the palate can be involved, resulting in tumefactive or erosive lesions [59]. Although the involvement of the oral cavity is an infrequent manifestation of IgG4-related disease, it should be taken into consideration as a possible differential diagnosis once neoplastic conditions are excluded. The histological examination represents the mainstay for the diagnosis of this condition [60].**Eosinophilic granulomatosis with polyangiitis (EGPA),** formerly known as Churg-Strauss syndrome, is a rare condition, immune-mediated, and multisystemic disease characterised by BE and TE, late-onset asthma, and small-to-medium vessel vasculitis [61]. Oral lesions represent an uncommon presentation; however, ulcers involve the palate, tongue, and floor of the mouth [62,63]. Gingival bleeding and painful blisters on the tongue have been reported by some research groups. In all these cases, the unresponsiveness to local measures and the detection of granular or necrotic lesions exhibiting an eosinophilic inflammatory infiltrate on biopsy may guide the tentative diagnosis. Also, the salivary glands may be involved in granulomatous inflammation, resulting in a marked swelling [64]. Finally, another recent case report by Suzuki et al. [65] described swelling of the floor of the mouth and cervical soft tissue-mimicking IgG4-related disease as an initial manifestation of EGPA.

## 3. Role of Eosinophils in Oral Precancerous Lesions

Oral potentially malignant disorders (OPMDs) include a group of conditions that have a higher risk of transformation to malignant lesions [66]. The most common forms of oral potentially malignant disorders (OPMDs) are erythroplakia, leukoplakia, lichen planus, and submucous fibrosis [67]. We conducted a thorough search using the PubMed database to identify pertinent articles relating to this early premalignant stage with reference to eosinophils, which yielded multiple human studies (Table 2). Overall, these studies suggest that OPMD samples exhibit a significant alteration of tissue eosinophils and a predominantly elevated tissue eosinophil count (TEC) compared to normal mucosa, which may correlate with the severity of dysplasia. Therefore, incorporating TEC diagnostic protocols may offer additional insights into the early transformation of dysplastic lesions to OSCC [68,69]. Supporting this theory, tumour-associated tissue eosinophilia (TATE) and tumour-associated blood eosinophilia (TABE) were identified in one study to be independent prognostic markers for premalignant and malignant OSCC, with elevated tissue eosinophils correlating with a favourable pre-malignancy prognosis, i.e., leukoplakia followed by dysplasia, respectively, while TABE was associated with poor prognosis [70]. Moreover, eosinophils may be considered as an indicator of invasion in oral intraepithelial neoplasia (OIN), and their presence may assist in cases of difficult diagnosis [71].

## 4. Role of Eosinophils in Oral Cancer

Eosinophilia is associated with multiple solid tumours, such as breast [74], colon [75], prostate [76], non-small cell lung cancer [77], oesophageal carcinoma [78], oral squamous cell carcinomas [79] and also haematological tumours, such as Hodgkin’s lymphoma [80]. Prolonged low-grade inflammation is often a prelude to cancer, and this is also the case with OSCC [81]. Tumour-associated tissue eosinophilia (TATE) is defined as “eosinophilic stromal infiltration of a tumour not associated with tumour necrosis or ulceration” and was first described in the carcinoma of the cervix in 1896 [82]. Currently, blood eosinophilia is regarded as a prognostic marker in malignant tumours with poor prognosis [83]. In the same manner, tissue eosinophilia at the site of tumour formation has been reported in various cancers, including the head and neck [84,85], of which oral cancer is a major subgroup, and other solid cancers [6].

The recruitment of eosinophils into the oral tumour site is mediated by inflammatory cytokines and chemokines, primarily attributed to IL-4 and IL-13, which are secreted from T helper cells (Th2) [86]. Eosinophils may promote tumour angiogenesis by the secretion of angiogenic factors, including the vascular endothelial growth factor (VEGF) and fibroblast growth factor-2 (FGF-2), granulocyte-macrophage colony-stimulating factor (GMCSF), and IL-8 [87,88,89]. Eosinophils can infiltrate tumour tissue and, in many cases, regulate their progression, such as interacting with tumour cells themselves by synthesising and secreting proteins, such as cytotoxic cationic proteins, including the eosinophil cationic protein (ECP), major basic protein (MBP), eosinophil peroxidise (EPO), granzyme and TNF, which may result in their death [79]. Eosinophils can also modify the microenvironment by secreting tumour-promoting mediators, such as pro-angiogenic factors and matrix-remodelling soluble mediators (such as matrix metalloproteinases (MMPs)) depending on the tumour microenvironment [90] The role of eosinophils in cancer biology and invasion is complex, with these cells have both pro- and anti-tumourigenic properties, possibly specific to the type of cancer. For example, in several solid tumours, such as gastric, colorectal, and prostate cancer, eosinophils appear to have an anti-tumourigenic role, while in others, such as cervical carcinoma, they correlate with poor clinical outcomes [6,90,91]. In yet other cancers, the role of eosinophils is undefined or plays no specific role. This would seem to suggest that the role of eosinophils could be cancer-dependent. Increasing evidence in experimental models indicates that modifying eosinophil behaviour/numbers could represent a new therapeutic strategy, particularly the depletion of eosinophils by targeting IL-5, which is the key cytokine for eosinophils proliferation, survival and priming, currently being assessed for eosinophilic asthma [15].

Interestingly, both TATE and blood eosinophilia can occur independently and have opposite effects on tumour behaviour; a study conducted on 150 cases of oral premalignant lesions and oral squamous cell carcinoma of varying grades observed that TATE exhibited a favourable prognosis. The maximum mean TEC was higher in premalignant conditions (i.e., Leucoplakia > dysplasia), followed by well, moderately, and poorly differentiated SCC, respectively. While an elevated TABE is associated with poor prognosis in high-grade OSCC and follow-up, these cases show early nodal metastasis and recurrence [70]. The main roles of eosinophils in oral cancer behaviour and biology have been graphically summarised (Figure 8).

Many studies have shown that elevated eosinophil infiltration is associated with a well-differentiated oral tumour compared to poorly differentiated ones, correlating with favourable prognosis and improved clinical TNM staging [70,84,85,92,93,94]. In contrast, other studies have shown that higher TATE is linked to poor prognosis and tumour invasion [69,79,95,96,97]. The main findings from the latter studies are illustrated in Table 3. This controversy may be attributed to the contrasting roles and effects of eosinophils in the cancer microenvironment. One study demonstrated that by inhibiting IL-5, a crucial cytokine for eosinophil activation, led to a reduction in tumour formation in a hamster OSCC model, revealing a major role for eosinophils in cancer development [98]. Eosinophils also have anti-tumour effects via the secretion of various substances (TNF-α, granzyme, cationic proteins, and IL-18) [6]. A study involving 24 cases of OSCC showed that a higher TATE was associated with OSCC compared to normal tissue. This study also showed that higher levels of TATE occurred in well-differentiated OSCC cases compared to other grades, and hence, TATE appears to be associated with favourable OSSC prognosis [94].

## 5. Discussion

Eosinophils are important innate effector cells that protect the host against invading pathogens by releasing preformed toxic granular mediators and producing reactive oxygen species to kill these invaders. In addition, eosinophils play a diverse and as yet undetermined role in modulating tumourigenesis. The detection of eosinophils in histopathological examination is straightforward since intact eosinophils are readily detected by routine hematoxylin and eosin (H&E) staining. Additional methods can be utilised, such as immunohistochemistry or autofluorescence, to specifically identify intact or degranulated eosinophils [83]. Specialist dyes such as Congo red stain (sodium salt of 3,3′-([1,1′-biphenyl]-4,4′-diyl)bis(4-aminonaphthalene-1-sulfonic acid)) have the ability to specifically bind to eosinophils [69]; this is particularly important in the diagnosis of OSSC, where biopsy specimens are limiting. A recent OSSC study found that the staining efficiency of Congo red stain over H&E staining to differentiate eosinophils was significant, with eosinophil infiltration observed in 86% of OSCC cases, although no significant correlation was found with the OSCC grade [103]. Congo red stain, therefore, provides a useful adjunct to H&E staining in OSCC. With advances in technology, flow cytometry has been utilised for immune cell profiling, including eosinophils in both human blood and tissue according to their CD11b/CD62L surface markers in addition to cytokine surface markers associated with differentiation (e.g., IL-5Ra/CD125 and IL-3Ra/CD123) [104,105]. Employing a flow cytometry approach to fully enumerate and characterise neutrophil differentiation stages enables a more accurate assessment of tumour-associated tissue eosinophilia (TATE) in oral cancer samples.

Eosinophils are associated with oral autoimmune diseases, such as BP, inflammatory and immune-mediated disorders, such as Kimura’s disease, EGPA, and HES, all of which are rare in the oral cavity. However, it is important to highlight that the current reporting of the association of eosinophils with oral diseases is inadequately documented in the existing literature.

Regular check-ups are essential for oral potentially malignant lesions to monitor for signs of malignant transformation. Researchers persist in their endeavours to discover a biomarker capable of predicting such transformations. We identified several studies in the literature that utilised TEC in oral premalignant lesions to assess lesion prognosis. The predominant findings from these studies indicate that a finding of higher eosinophil infiltration in pre-malignant lesions should prompt a detailed investigation to rule out the invasion. Researchers have also suggested that TEC may be used as an adjunct tool in cases of suspected malignant transformation in oral leukoplakia with dysplastic changes [68,69,71]. Furthermore, an interesting study by Reddy et al. revealed that, among 30 cases of oral submucosal fibrosis, 57% exhibited peripheral blood eosinophilia [106].

The role of eosinophils in oral cancer remains an enigma, and their presence may be attributed to substances secreted by the cancer itself, which attract eosinophils to the tumour site [107]. Similarly, the recruitment of eosinophils into the cancer site may be initiated by chemokines produced and secreted by Th2 cells [72]. Controversy remains in the literature regarding the effects of eosinophils in oral cancer behaviour and invasion, thus limiting its use in routine diagnosis. These contrasting regulatory roles for eosinophils in OSCC may account for the broad range of soluble mediators they secrete in addition to cytokines, such as IL-4 and IL-13, which trigger Th2 cell differentiation and B cell activation. For instance, eosinophils can exhibit pro-tumour effects; IL-4 has the potential to facilitate tumour growth due to its anti-apoptotic properties [108] whilst also having anti-angiogenic properties that inhibit tumour growth [109]. Similarly, IL-13 plays a role in anti-tumour immune responses; however, it may inhibit anti-tumour immunity by suppressing IFN-*γ* secretion, which promotes Th1 responses and CD8^+^ cytotoxic T lymphocyte activity [110]. Overall, eosinophils have multiple complex roles to play in tumourigenesis, particularly in OSCC, and the main findings are illustrated in Table 2. There are multiple mechanisms for eosinophil homing to the tumour microenvironment, where they can potentially kill tumour cells but also release factors that promote tumourigenesis. The use of eosinophils as predictive markers for oral pre-malignancy and OSCC is emerging but requires further clinical research.

While acknowledging the crucial role of eosinophils in oral diseases, this review highlights several key limitations. These include a scarcity of documented evidence linking eosinophils to various oral diseases, controversies surrounding their role in oral cancer behaviour, and a notable absence of studies investigating their involvement in oral candidiasis and viral infections. Additionally, uncertainties persist regarding the predictive value of tissue eosinophilia in oral potentially malignant disorders and cancer.

## 6. Limitations and Future Implications

While acknowledging the crucial role of eosinophils in oral diseases, this review highlights several key limitations and future research directions. Despite extensive studies elucidating the role of eosinophils in fungal infections, particularly candida albicans [111], a comprehensive search using databases, such as Medline via Ovid and Embase, along with manual searches on Google Scholar, failed to retrieve any studies linking eosinophils to oral candidiasis. Similarly, while several studies have discussed the role of eosinophils in viral infections, particularly respiratory viral infections [112], such as the Herpes Simplex Infection, Varicella Zoster Infection, Herpes Zoster Infection, and Epstein–Barr Virus Infection, no specific articles relating to oral viral infections were found through systematic searches. Additionally, controversies persist regarding the involvement of eosinophils in oral cancer behaviour, and uncertainties surround the predictive value of tissue eosinophilia in oral potentially malignant disorders and cancer. These gaps underscore the need for further research to clarify the role of eosinophils in oral pathologies and improve diagnostic and prognostic strategies.

## Figures and Tables

**Figure 2 ijms-25-04373-f002:**
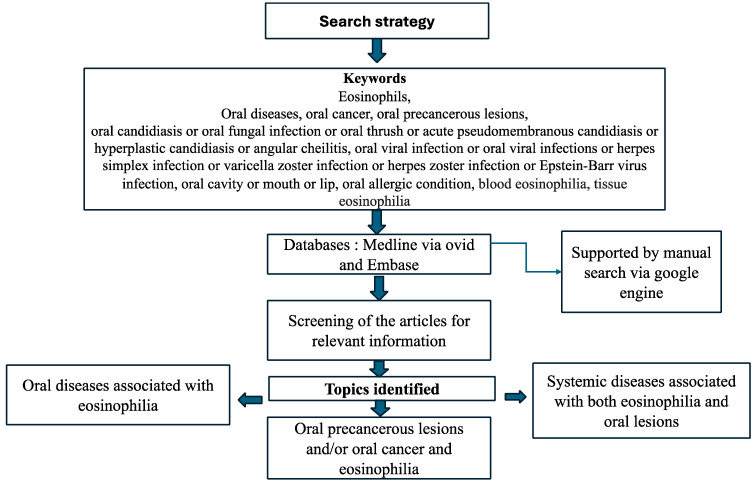
Schematic representation of our search structure. This diagram is an original creation drawn by the authors.

**Figure 3 ijms-25-04373-f003:**
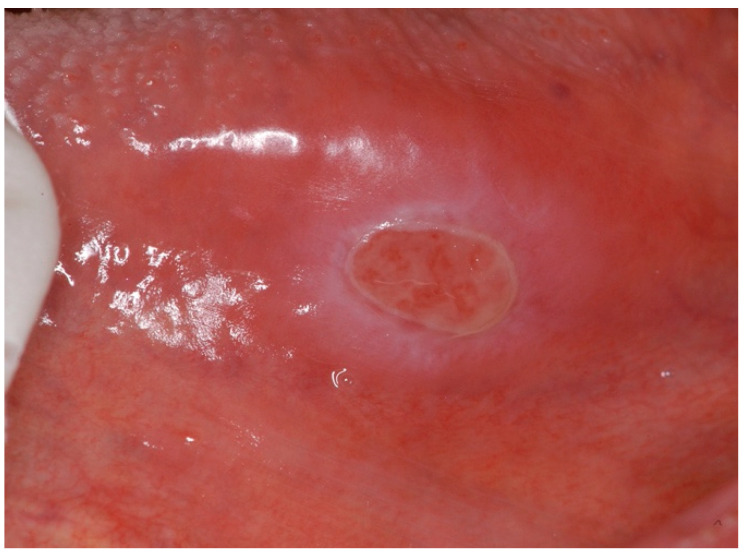
Clinical features of an oral eosinophilic ulcer on the left border of the tongue extending to the ventral area (Courtesy of A/Prof. Antonio Celentano, Melbourne Dental School, The University of Melbourne, Australia. This photograph is an original creation. All rights reserved).

**Figure 4 ijms-25-04373-f004:**
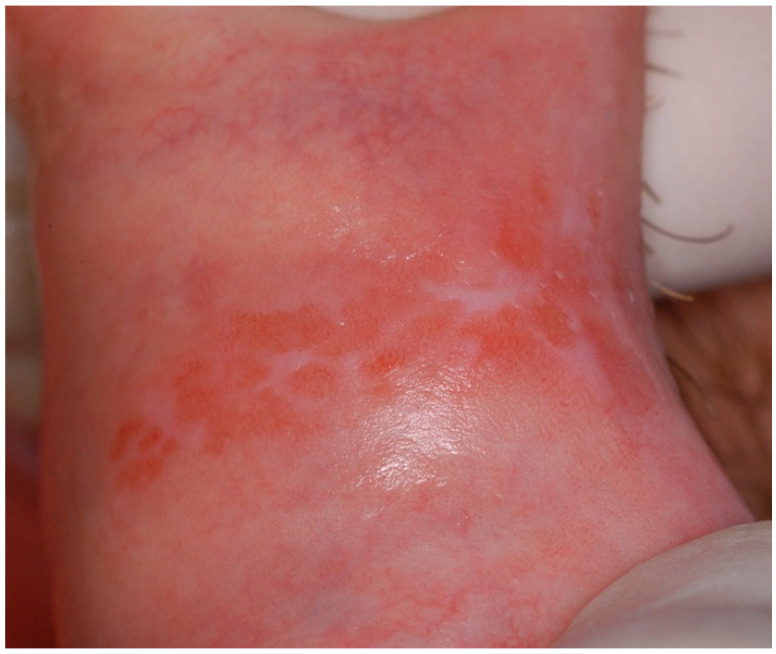
Lichenoid leukokeratosis of the left retrocommissural area with mild intraepithelial dyskeratosis (courtesy of A/Prof. Antonio Celentano, Melbourne Dental School, The University of Melbourne, Australia. This photograph is an original creation. All rights reserved).

**Figure 5 ijms-25-04373-f005:**
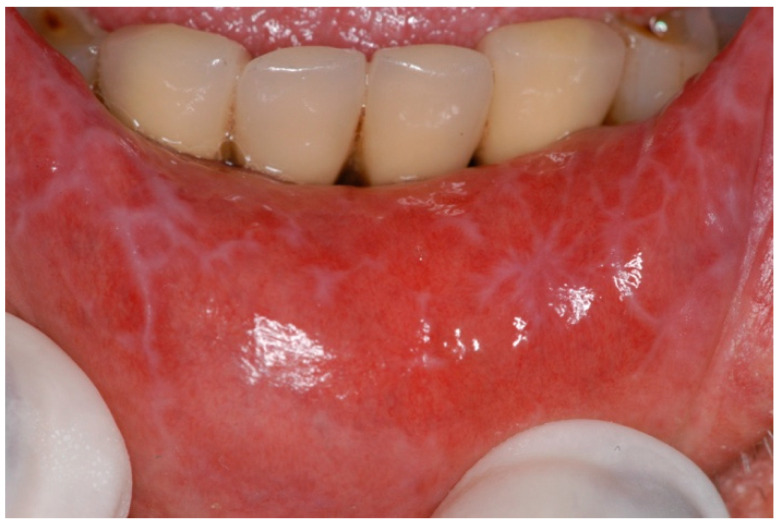
Oral chronic graft versus host disease: widespread lichenoid lesions involving the lower labial mucosa and vermilion border of a patient with acute myeloid leukaemia who underwent allogeneic cell transplant (courtesy of A/Prof. Antonio Celentano, Melbourne Dental School, The University of Melbourne, Australia. This photograph is an original creation. All rights reserved).

**Figure 6 ijms-25-04373-f006:**
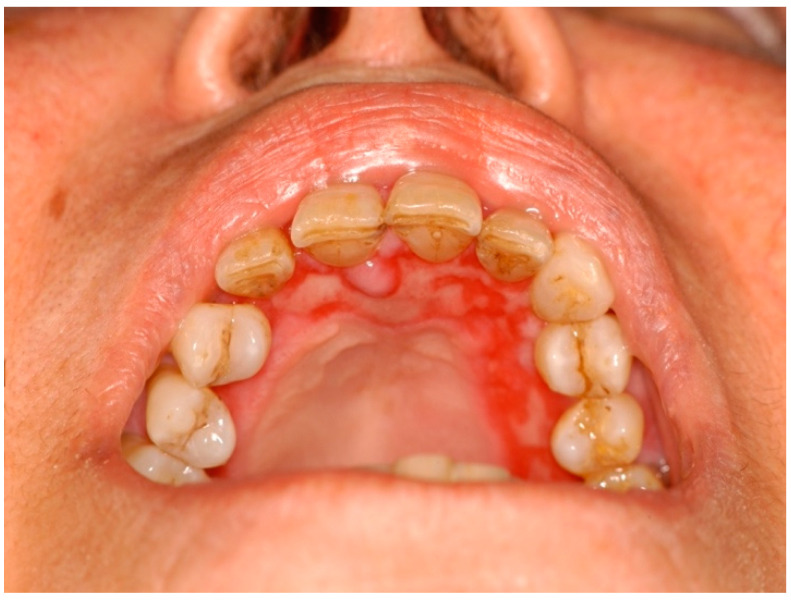
Bullous pemphigoid with oral onset, presenting with lesions involving the entire upper alveolar mucosa (courtesy of A/Prof. Antonio Celentano, Melbourne Dental School, The University of Melbourne, Australia. This photograph is an original creation. All rights reserved).

**Figure 7 ijms-25-04373-f007:**
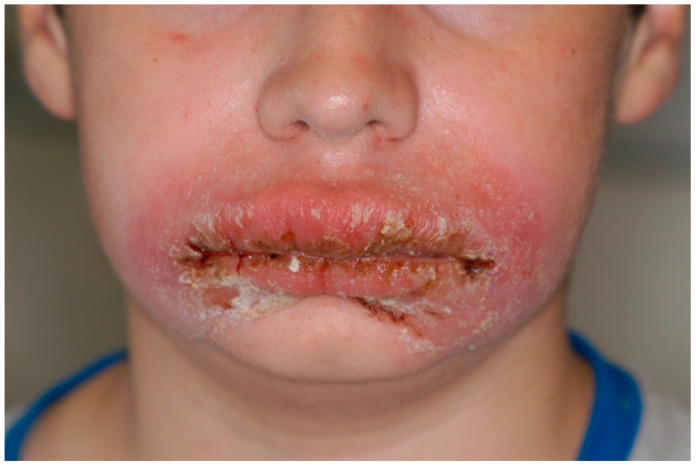
Oral Crohn’s in a paediatric patient presenting with diffuse swelling of the lower third of the face, gross enlargement of the upper and lower lips, and linear ulcers (courtesy of A/Prof. Antonio Celentano, Melbourne Dental School, The University of Melbourne, Australia. This photograph is an original creation. All rights reserved).

**Figure 8 ijms-25-04373-f008:**
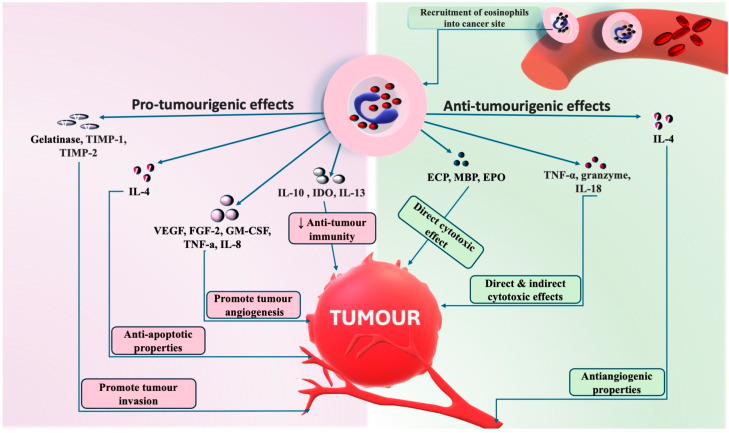
Pro-tumourigenic and anti-tumourigenic effects of eosinophils in oral cancer. Pro-tumourigenic activities associated with eosinophils include the activation of gelatinase, which aids in basement membrane degradation and facilitates tumour invasion. Eosinophils also release pre-formed matrix metalloproteinases (MMPs), such as MMP-9 and inhibitory molecules (TIMP-1 and TIMP-2) that participate in extracellular matrix remodelling. Additionally, eosinophils promote tumour angiogenesis by secreting angiogenic factors, including the vascular endothelial growth factor (VEGF), fibroblast growth factor-2 (FGF-2), tumour necrosis factor (TNF), granulocyte-macrophage colony-stimulating factor (GM-CSF), and IL-8. Eosinophils may also contribute to the downregulation of anti-tumour immunity through the secretion of cytokines, such as IL-10 and indoleamine oxidase (IDO). Conversely, eosinophils exhibit anti-tumourigenic effects directly mediated by the killing of tumour cells via eosinophil cationic protein (ECP), major basic protein (MBP), and eosinophil peroxidase (EPO). The release of TNF-a, IL-18, and other cytokines contributes to tumour cytotoxicity either directly or indirectly by stimulating additional effector cells. This figure is an original creation drawn by the authors. All rights reserved.

**Table 1 ijms-25-04373-t001:** Systemic diseases with oral manifestations that are associated with TE and/or BE.

Systemic Diseases Associated with Oral Lesions and Eosinophilia	
Disease	Oral Manifestations
Crohn’s disease (cd)	Ulcers, fissuring of the lip, cobblestoning of the buccal mucosa, angular cheilitis, mucosal polyps and perioral erythema
Kimura’s disease	A non-tender subcutaneous lesion in the head and neck region
Hypereosinophilic syndrome (HES)	Painful mucosal lesions presenting as ulcerations and erosions localised to the lips, gingiva, tongue, and palatal mucosa
IgG4-related disease	Tumefactive or erosive lesions on the tongue or palate
Eosinophilic granulomatosis with polyangiitis (EGPA)	Ulcers involving the palate, tongue, and floor of the mouth, gingival bleeding, and tongue blisters

**Table 2 ijms-25-04373-t002:** Human studies investigating the role of eosinophils in oral potentially malignant disorders (OPMDs).

Study	Premalignant Lesion Sample Size (n): (Type)	Method of Counting Eosinophils	Main Findings	Probability (*p*-Value)
Jain et al., 2014 [72]	20: (dysplasia)	Congo red stain	No significant difference in the degree of dysplasia and eos count.	(*p* = 0.652).
Madhura et al., 2015 [68]	59: (leukoplakia)	H&E	↑ TEC in oral leukoplakia compared to normal tissues. Oral leukoplakia versus control	(*p* = 0.001).
Kargahi et al., 2015 [73]	20: (dysplasia)	H&E, Congo red stain and IHC	↑ Number of eosinophils in dysplastic mucosa compared to normal mucosa and in OSCC compared to dysplastic mucosa	(*p* < 0.001).
Martinelli-Kläy et al., 2018 [71]	16: (OIN-1), 18: (OIN-2), 17: (OIN-3)	H&E	Eos distribution is associated with diagnosis severity	(*p* < 0.01).
Deepthi et al., 2019 [69]	20: ( OED and oral leukoplakia)	Congo red stain	TATE may play a role in stromal invasion	(*p* < 0.05).
Kaur et al., 2023 [70]	38: (leukoplakia) 32: (dysplasia)	H&E	TATE with TABE are independent prognostic markers in premalignant and malignant OSCC. ↑ tissue eos is a favourable pre-malignancy prognosis; ↑ TABE is a poor prognosis in high-grade OSCC.	(*p* < 0.001).

Abbreviation list: Eos = eosinophils; H&E = hematoxylin and eosin; IHC = immunohistochemistry; OSCC = oral squamous cell carcinoma; OED = oral epithelial dysplasia; OIN = oral intraepithelial neoplasia; TATE = tumour-associated tissue eosinophilia; TABE = tumour-associated blood eosinophilia; TEC = total eosinophil counts.

**Table 3 ijms-25-04373-t003:** Human studies investigating the role of eosinophils in OSCC.

Study	Oral Cancer Sample Size (n)	Method of Diagnosis	Main Findings	Probability (*p*-Value)
Lowe and Fletcher, 1984 [85]	136	N/R	↑ tissue eosinophilia is related to tumour histological differentiation and is associated with favourable prognosis. ↑ Circulating eosinophilia is associated with metastatic disease and poor prognosis.	(*p* value; N/R).
Goldsmith et al., 1987 [93]	16	H&E	Stromal eosinophilia is significantly correlated with a favourable outcome.	(*p* = 0.018).
Goldsmith et al., 1992 [84]	51	H&E	↑eosinophilia in HNSCC stroma is a favourable prognostic indicator. Furthermore, high-grade TATE may have a protective effect on the development of distant metastasis.	(*p* < 0.001), (*p* < 0.05), respectively.
Horiuchi et al., 1993 [97]	31	H&E	↑ eos infiltration and the expression of the HLA-DR antigen on tumour cells shows an unfavourable prognosis.	(*p* < 0.05).
Dorta et al., 2002 [92]	125	Morphometric analysis. (a 25-point ocular graticle under 800 magnification)	Blood eosinophilia was present in 34.4% of cases; tissue eosinophilia was present in 45.6% of cases. ↑ TATE is a possible favourable prognostic factor in OSCC clinical TNM stages II/III.	(*p* = 0.015).
Alrawi et al., 2005 [95]	4	H&E	↑ TEC is a histopathologic marker associated with tumour invasion and is a clinical predictor for aggressive tumourigenesis.	(*p* < 0.005).
Falconieri et al., 2008 [96]	13	H&E	↑ eos infiltrate is associated with stromal invasion in OSCC.	(*p* value; N/R).
Tostes Oliveira et al., 2009 [99]	43	H&E	No statistically significant association between ↑ TATE and muscular infiltration in OSCC. The close relationship between eosinophils and striated muscular fibre damage is frequently observed; this suggests that ↑ TATE is associated with OSCC invasiveness. Overall survival and disease-free survival rates were equivalent for both OSCC with intense and absent/mild tissue eosinophilia.	(*p* = 0.009).
Jain et al., 2014 [72]	40	Congo red stain	↑ TEC in OSCC compared to dysplasia suggests its role in the stromal invasion; non-metastatic cases showed ↑ eso counts more than metastatic carcinomas. Eosinophilia showed a favourable histopathological prognostic factor in OSCC.	*p* < 0.0001, (*p* < 0.0001), respectively
Sahni et al., 2015 [94]	24	Congo red stain	↑ eos infiltration in the well-differentiated lesions compared to the lower grades. Eosinophils play a positive role in circumventing tumour invasion	(*p* = 0.006).
Kargahi et al., 2015 [73]	20	H&E, Congo red, and IHC	↑ TEC in dysplastic mucosa compared to normal mucosa.	(*p* < 0.001).
Rakesh et al., 2015 [100]	14	H&E	↑ TATE was significantly associated with Loco regional recurrence.	(*p* < 0.001).
Debta et al., 2016 [101]	30	H&E	Among 30 cases of OSCC, 63.33% were TATE^+^, and 36.66% were TATE^-^. Eosinophil infiltrates ↓, from tumour Stage 1 to Stage 3 and ↓ from well to poorly differentiated carcinoma.	(*p* < 0.05).
Martinelli-Kläy et al., 2018 [71]	32	H&E	The distribution of eosinophils per 10 hpf was significantly associated with the severity of the diagnosis. Moreover, although not significantly different, non-metastatic invasive OSCC had a higher number of cases (68.2%) with ≥22 eos/10 hpf contrasting with 50% in metastatic OSCC	(*p* < 0.01).
Peurala et al., 2018 [20]	83 (oral cavity); 16 (lip SCC)	H&E	↑ TATE showed significantly better survival than ↓ TATE. TATE is a prognostic marker for oral and lip SCC: more than 4 eosinophils/HPF may predict a more favourable prognosis	(*p* = 0.0136).
Deepthi et al., 2019 [69]	50	Congo red stain	↑ TEC in OSCC compared to OED. Mean TEC = 2.12 in OED and 4.31 in OSCC. ↑ TEC is a poor prognosis in OSCC.	(*p* = 0.000026).
Siddiqui et al., 2020 [79]	30	H&E	↑ TATE value associated with poorly differentiated carcinoma. Statistically significant correlation between TATE and OSCC histological grade. Eosinophilia of the peripheral blood = adverse sign in OSCC patients	(*p* ≤ 0.001).
Sethi et al., 2020 [102]	60	IHC (anti-CD15 ab)	Eosinophil count correlates with tumour differentiation. Quantification did not correlate with clinical staging. The mean numbers of eosinophils in well to moderately differentiated OSCC = 15.37 ± 11.86. The mean numbers of eosinophils in poorly differentiated OSCC = 12.62 ± 14.30.	(*p*; no access to full text article)
Kaur et al., 2023 [70]	80	H&E	↑ TATE showed a favourable prognosis; ↑ TATE in premalignant conditions (leucoplakia > dysplasia) was followed by WDSCC, MDSCC, and PDSCC, respectively. ↑ TABE shows poor prognosis in high-grade OSCC and follow-ups.	(*p* < 0.001).

Abbreviation list: ab = antibody; eos = eosinophils; H&E = hematoxylin and eosin; HLA-DR = human leukocyte antigen; HPF = high-power field; HNSCC = head and neck squamous carcinoma; IHC = immunohistochemistry; MDSCC = moderately differentiated squamous cell carcinoma; N/R = not reported; OSCC = oral squamous cell carcinoma; OED = oral epithelial dysplasia; PDSCC = poorly differentiated squamous cell carcinoma; TATE = tumour-associated tissue eosinophilia; TABE = tumour-associated blood eosinophilia; TEC = total eosinophil counts; WDSCC = well-differentiated squamous cell carcinoma.

## Data Availability

The data that support the findings of this study are available from the corresponding author upon reasonable request.

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
