# Peer review of "Eosinophils in Oral Disease: A Narrative Review"

_ijms, 2024, doi:10.3390/ijms25084373_

Round 1

Reviewer 1 Report

Comments and Suggestions for Authors

Dear Authors,

Thank You for a pleasure to read Your work.

I have several comments and offers to improve Your article.

The title is rather interesting for social media but not for scientific journal with high ranking. Please, change it to more appropriate with point to the type (narrative review, systematic review or other).

Abstract

We understand that You try to set the eosinophils importance, but it is better to correct the introduction without so often use of this word.

Introduction

Please, write in the end of introduction the aim of Your study.

Role of Eosinophils in oral precancerous lesions will be better to replace to the oral diseases not with systematic diseases.

Please, make the figure 1 more readable and point who is owner of it.

Please, write for tables 1 and 2 probabilities.

You have the greatest part of article for eosinophils and precancer/cancer connection statement but for other diseases as systematic You tell briefly. Please, re-write article according to common direction as it was in the beginning or leave only precancer/cancer.

Also, there is no section ‘Discussion’. Please, add it and write the limitation of your study in the end.

Sincerely, Reviewer

Author Response

Dear Editor,

Re: “Orchestrators of oral immunity: the fascinating role of eosinophils”, now re-titled:

Eosinophils in oral disease: a narrative review

By

Huda Moutaz Asmael Al-Azzawi, Rita Paolini, Nicola Cirillo, Lorraine A. O’ Reill, Ilaria Mormile, Caroline Moore, Tami Yap, Antonio Celentano

We would be most grateful if you would consider the revised version of the above manuscript for publication in the International Journal of Molecular Sciences, within the Special Issue “Eosinophils: Current Status and Future Perspective in Health and Disease”.

The manuscript has been extensively revised using track changes to address the feedback received from the reviewers. The corrections made (point by point) are also available at the bottom of this letter for ease of reference.

We would like to thank you once again for your most appreciated consideration. Additionally, we would like to thank the reviewers for their constructive and positive comments.

Yours sincerely,

A/Prof Antonio Celentano

[email protected]

Ph: +610435836651

Reviewer 1

Comment 1: Dear Authors, Thank You for a pleasure to read Your work. I have several comments and offers to improve Your article.

Reply: We appreciate the reviewer found our article interesting and will promptly address all the constructive comments.

Comment 2: The title is rather interesting for social media but not for scientific journal with high ranking. Please, change it to more appropriate with point to the type (narrative review, systematic review or other)..

Reply: We appreciate the reviewer's comment. We updated our article title to: “Eosinophils in oral disease: a narrative review” accordingly.

Comment 3: Abstract: We understand that You try to set the eosinophils importance, but it is better to correct the introduction without so often use of this word.

Reply: We appreciate the reviewer's comment. As per your suggestion, we have reduced the repetition of the word 'eosinophils' throughout the abstract text, as much as we could.

Comment 4: Introduction: Please, write in the end of introduction the aim of Your study.

Reply: Thank you for highlighting your concern. We've revised the manuscript introduction section to include the aim of this review article at the end, this added paragraph now reads: “The aim of this study is to provide a comprehensive overview of the role of eosinophils in oral diseases, including their involvement in tissue eosinophilia, blood eosinophilia, and systemic diseases with oral manifestations, aiming to enhance understanding among clinicians and pathologists and to serve as a reference for diagnosis and management in oral medicine”.

Comment 5: Role of Eosinophils in oral precancerous lesions will be better to replace to the oral diseases not with systematic diseases.

Reply: Thank you for your comment. While we respect this reviewer’s opinion, we would like to bring to your attention that our original decision to stratify the diseases in this manner was reached after thorough consultation among a group of experts in oral medicine and pathology. We decided to structure this into two preliminary paragraphs: “Oral diseases - associated with TE and/or blood eosinophilia (BE)” and “Systemic diseases - associated with both eosinophilia and oral lesions.” Additionally, as is often done in several seminal review articles clinically oriented, we allocated dedicated, separate sections for precancer and cancer. This approach has several beneficial aspects for authors, as well as for students/trainees or clinicians who would like rapid consultation on these two specific areas. We acknowledge that our structure may not have been properly highlighted at the beginning of our manuscript, so we slightly corrected the conclusive portion of the introduction section to better clarify this aspect, while still retaining our decided manuscript structure. This now reads: “We present the current knowledge base in this subject area to provide a more comprehensive understanding of the relationship between eosinophils and oral diseases, which have been stratified here into two main categories:

  1. Oral diseases - associated with TE and/or blood eosinophilia (BE).
  2. Systemic diseases - associated with both eosinophilia and oral lesions.

These are followed by two additional paragraphs dedicated to exploring the role of eosinophils in oral precancerous lesions and oral cancer.”

Thank you for your understanding.

Comment 6: Please, make the figure 1 more readable and point who is owner of it.

Reply: We thank the reviewer for their comment. This figure, now relabelled “Figure 3” has been resized to the maximum allowed by the Journal template, resulting in improved readability meeting acceptable standards. Regarding the ownership of this figure, we confirm that it is an original creation drawn from scratch by the authors. Following your suggestion, this clarification has been added to the figure legend “Figure 3: Pro-tumourigenic and anti-tumourigenic effects of eosinophils in oral cancer. Pro-tumorigenic activities associated with eosinophils include the activation of gelatinase, […] This figure is an original creation drawn by the authors”.

Comment 7: Please, write for tables 1 and 2 probabilities.

Reply: We thank the reviewer for their insight. We have made adjustments within Table 1 and Table 2 to include p-values from all the included studies where these were available.

Comment 8: You have the greatest part of article for eosinophils and precancer/cancer connection statement but for other diseases as systematic You tell briefly. Please, re-write article according to common direction as it was in the beginning or leave only precancer/cancer.

Reply: We sincerely appreciate your insightful comment. While we hold great respect for the reviewer’s perspective, we must emphasize that confining our entire review solely to the precancerous and cancerous aspects is not aligned with the original intention of our work. However, we acknowledge the reviewer's observation that our emphasis on precancer and cancer may have been more pronounced during our initial discussions. This emphasis stemmed from the recognition that while the link between eosinophils and most oral/systemic disorders has been extensively explored, the association with precancerous and cancerous conditions remains subject to significant debate and conflicting evidence. Therefore, we deliberately highlighted this aspect to stimulate critical clinical and translational discourse. Additionally, it's worth noting that while oral/systemic disorders were treated separately with concise, point-by-point paragraphs, the categories of cancer and particularly pre-cancer encompass a vast number of different pathological entities, thus necessitating increased dedication of space. We hope this clarification underscores our rationale for the allocation of text to specific manuscript areas.

Comment 9: Also, there is no section ‘Discussion’. Please, add it and write the limitation of your study in the end. Sincerely, Reviewer

Reply: Thank you for your valuable feedback. We appreciate your thorough review of our manuscript. We would like to kindly point out that a 'Discussion' section does indeed exist in our manuscript, located immediately after Table 2, at the beginning of page 15. In the 'Discussion' section, we have elaborated on the limitations of our study. However, we understand that it may have been overlooked. To address this, we have now condensed the originally drafted “Future directions” section with an additional limitation paragraph, which now reads “Limitations and future implications” at the end of our manuscript, to give it more prominence in this revised version.

Reviewer 2 Report

Comments and Suggestions for Authors

Dear authors, thank you for submitting the manuscript "Orchestrators of oral immunity: the fascinating role of eosinophils"

-Please include in the title what type of review is it (narrative).

-Minor English revision is needed, I see some sentences that need a few commas.

-Your manuscript will be more interesting to readers if you provide an image of an eosinohil (example transmission electron microscopy or at least a drawing).

-Include clinical images whenever you mention those oral diseases associated with tissue eosiniphilia and/or blood eosinohilia.

-Make a small table summarizing all the diseases described in 2.2

-Create a paragraph mentioning the limitations of your narrative review (example making a systematic review, do the search in more languages, etc)

-Include a graphic abstract summarizing your search and its sequence.

Comments on the Quality of English Language

minor revision is recommended

Author Response

Dear Editor,

Re: “Orchestrators of oral immunity: the fascinating role of eosinophils”, now re-titled:

Eosinophils in oral disease: a narrative review

By

Huda Moutaz Asmael Al-Azzawi, Rita Paolini, Nicola Cirillo, Lorraine A. O’ Reill, Ilaria Mormile, Caroline Moore, Tami Yap, Antonio Celentano

We would be most grateful if you would consider the revised version of the above manuscript for publication in the International Journal of Molecular Sciences, within the Special Issue “Eosinophils: Current Status and Future Perspective in Health and Disease”.

The manuscript has been extensively revised using track changes to address the feedback received from the reviewers. The corrections made (point by point) are also available at the bottom of this letter for ease of reference.

We would like to thank you once again for your most appreciated consideration. Additionally, we would like to thank the reviewers for their constructive and positive comments.

Yours sincerely,

A/Prof Antonio Celentano

[email protected]

Ph: +610435836651

Reviewer 2

Comment 1: Dear authors, thank you for submitting the manuscript "Orchestrators of oral immunity: the fascinating role of eosinophils" -Please include in the title what type of review is it (narrative).

Reply: Thank you for your thoughtful comment. Considering also the feedback from the reviewer 1, who suggested a similar change, we have updated our article title to: “Eosinophils in Oral Disease: A Narrative Review” accordingly.

Comment 2: Minor English revision is needed, I see some sentences that need a few commas.

Reply: Thank you for your comment. The manuscript has now undergone four cycles of revision by four native English language speakers, aimed at intercepting any style and punctuation issues. We hope this reviewer find the revised version more readable,

Comment 3: Your manuscript will be more interesting to readers if you provide an image of an eosinophil (example transmission electron microscopy or at least a drawing).

Reply: Thank you for your comment. As per your suggestion, we have added an appealing and schematic iconography of an eosinophil retrieved from a license-free collection (By BruceBlaus). When using this image in external sources, it can be cited as: Blausen.com staff (2014). "Medical gallery of Blausen Medical 2014". WikiJournal of Medicine 1 (2). DOI:10.15347/wjm/2014.010. ISSN 2002-4436. - Own work, CC BY 3.0, https://commons.wikimedia.org/w/index.php?curid=28223981), quoting the original source as per their guidelines. This now reads as Figure 1.

Comment 4: Include clinical images whenever you mention those oral diseases associated with tissue eosiniphilia and/or blood eosinohilia.

Reply: Thank you for your comment. While it is virtually impossible for any author to procure original clinical pictures for every single condition described, these authors have made every effort to provide readers with a collection of remarkable original clinical photographs depicting some of the most prominent conditions described. These additional 5 high resolution clinical images are now embedded into the manuscript bringing the final number of figures for this manuscript to 8.

Comment 5: Make a small table summarizing all the diseases described in 2.2

Reply: Thank you for your comment. According to your suggestion, we have now included a small summarizing table to describe all diseases with oral manifestations included in section 2.2

Comment 6: Create a paragraph mentioning the limitations of your narrative review (example making a systematic review, do the search in more languages, etc)

Reply: Thank you for your thoughtful comment. Considering also the feedback from the reviewer 1, who suggested a similar change, we have updated our article to include a final “Limitations and future implications” paragraph at the end of our manuscript, to give limitations more prominence in this revised version.

Comment 7: Include a graphic abstract summarizing your search and its sequence.

Reply: We appreciate the reviewer's comment. While we do not feel confident in including a proper prisma chart-like diagram given the non-systematic nature of this review, we still tried to address this reviewer/s comment creating a very concise summative flow chart of the search conducted. This is labelled as Figure 2

Round 2

Reviewer 1 Report

Comments and Suggestions for Authors

Dear Authors,

Thank You for Your corrections.

The track system did not show me any changes but I found them myself according to Your step-by-step answers. 

Please, add the known probabilites for studies in SEPARATE column as it will be more readable. 

Please, add for all photos and schemes the same statement that is your own.

For systematic diseases, please, add in the text explanation that is closer to that was in your response for me.

Sincerely, Reviewer 

Author Response

Dear Editor,

Re: “Eosinophils in oral disease: a narrative review

By

Huda Moutaz Asmael Al-Azzawi, Rita Paolini, Nicola Cirillo, Lorraine A. O’ Reill, Ilaria Mormile, Caroline Moore, Tami Yap, Antonio Celentano

We would be most grateful if you would consider the revised version of the above manuscript for publication in the International Journal of Molecular Sciences, within the Special Issue “Eosinophils: Current Status and Future Perspective in Health and Disease”.

The manuscript has been Revised using track changes to address the feedback received from the reviewers. The corrections made (point by point) are also available at the bottom of this letter for ease of reference.

We would like to thank you once again for your most appreciated consideration. Additionally, we would like to thank the reviewers for their constructive and positive comments.

Yours sincerely,

A/Prof Antonio Celentano

[email protected]

Ph: +610435836651

Reviewer 1

Dear Authors, Thank You for Your corrections. The track system did not show me any changes but I found them myself according to Your step-by-step answers.

Comment 1: Please, add the known probabilites for studies in SEPARATE column as it will be more readable.

Reply: Thank you for your feedback. We have incorporated all available probabilities for the studies in a dedicated column, as per your request.

Comment 2: Please, add for all photos and schemes the same statement that is your own.

Reply: Thank you for your comment. We have now included for all photographs and diagrams the requested statements.

Comment 3: For systematic diseases, please, add in the text explanation that is closer to that was in your response for me. Sincerely, Reviewer

Reply: "We value the reviewer's feedback and have incorporated an explanation on page 5, as requested: 'Our deliberate emphasis aimed to foster discussion on the debated link between eosinophils and these conditions. Moreover, the extensive range of pathological entities within cancer and precancer warranted greater space allocation compared to other oral/systemic disorders.'

We sincerely appreciate your thoughtful review, and we trust that we have adequately addressed all concerns raised during this review process.